



# On the short-term variability of turbulence and temperature in the winter mesosphere

Gerald A. Lehmacher[1], Miguel F. Larsen[1], Richard L. Collins[2], Aroh Barjatya[3], and Boris Strelnikov[4]

[1]Department of Physics & Astronomy, Clemson University, Clemson, South Carolina, USA
[2]Geophysical Institute, University of Alaska, Fairbanks, Alaska, USA
[3]Department of Physical Sciences, Embry-Riddle Aeronautical University, Daytona Beach, Florida, USA
[4]Leibniz-Institute for Atmospheric Physics, Kühlungsborn, Germany

**Correspondence:** Gerald A. Lehmacher (glehmac@clemson.edu)

**Abstract.** Four mesosphere-lower thermosphere temperature and turbulence profiles were obtained in situ within $\sim 30$ minutes and over an area of about 100 by 100 kilometers during a sounding rocket experiment conducted on January 26, 2015 at Poker Flat Research Range in Alaska. Using active payload attitude control, neutral density fluctuations, a tracer for turbulence, were observed with very little interference from the payload spin motion, and with high precision ($< 0.01\%$) at sub-meter

resolution. The large-scale vertical temperature structure was very consistent between the four soundings. The mesosphere was almost isothermal, which means more stratified, between 60 and 80 km, and again, between 88 and 95 km. The stratified regions adjoined quasi-adiabatic regions assumed to be well mixed. Additional evidence for vertical transport and convective activity comes from sodium densities and trimethyl aluminum trail development, respectively, which were both observed simultaneously with the in situ measurements. We found considerable kilometer scale temperature variability with amplitudes

of 20 K in the stratified region below 80 km. Several thin turbulent layers were embedded in this region, differing in width and altitude for each profile. Energy dissipation rates varied between 0.1 and 10 mW/kg, which is typical for the winter mesosphere. Very little turbulence was observed above 82 km, consistent with very weak small-scale gravity wave activity in the upper mesosphere during the launch night. On the other hand, above the cold and prominent mesopause at 102 km, large temperature excursions of +40 K to +70 K were observed. Simultaneous wind measurements revealed extreme wind shears near 108

15   km, and combined with the observed temperature gradient, isolated regions of unstable Richardson numbers ($0 < Ri < 0.25$) were detected in the lower thermosphere. The experiment was launched into a bright auroral arc under moderately disturbed conditions ($K_p \sim 5$).

## 1 Introduction

The structure and dynamics of the mesosphere are largely determined by atmospheric gravity waves (GWs) propagating from
the lower atmosphere (e.g., *Fritts and Alexander*, 2003). Large temperature and wind amplitudes lead to GW breaking, instabilities and intermittent turbulence. Such processes are too small to be included in global atmosphere models and must be parameterized with eddy diffusion coefficients. Large and variable eddy diffusion causes enhanced transport of minor species, e.g., O and NO, which in turn modify the structure and energy balance of the upper atmosphere (*Qian et al.*, 2009; *Meraner and*



*Schmidt*, 2016). GW interactions and breaking determine eddy heat flux, momentum flux divergence, mean flow acceleration, energy dissipation at viscous scales and the seeding of secondary GWs, which may propagate further into the thermosphere (e.g., *Snively et al.*, 2017).

The large variability of the northern winter mesosphere is well known (e.g., *Offermann*, 2009). Perturbations of the polar
vortex, stratospheric warmings, the formation of a planetary wave surf zone around 70 km, and the common occurrence of mesospheric inversion layers (MIL) (*Meriwether and Gerrard*, 2004) play a major role in this variability and the ability of GWs to reach the mesosphere. Measurements of turbulence in the northern winter mesosphere confirm the general increase of energy dissipation and eddy diffusion and demonstrate the large variability of temperature and associated turbulent structure (*Lübken et al.*, 1993; *Lübken* , 1997).

Modeling of mesospheric turbulence has advanced to multi-scale GW interactions; an example is the interaction of a small-scale GW with large scale MIL or with larger GWs (*Fritts et. al.*, 2018a, b, and references therein). Experimental studies with rocket borne ionization gauges have shown that small-scale turbulence is present in the very stable inversion layer, and in some cases, indicates mixing in the quasi-adiabatic layer above the inversion (*Lehmacher and Lübken*, 1995; *Lehmacher et al.*, 2006, 2011; *Szewczyk et al.*, 2013; *Strelnikov et al.*, 2017). Since rocket measurements of neutral turbulence are relatively complex
and costly, they often only provide a single profile of temperature and turbulence, while model results allow the analysis in the full spatial-temporal domain.

An early attempt at multi-point, in situ turbulence measurements was made by *Blix et al.* (1990) using a small spherical positive ion probe ejected from the main sounding rocket payload, which also carried a similar ion probe. Recently, *Strelnikov et al.* (2017) reported results using a payload with two ionization gauges at the front and back, for upleg and downleg neutral
density observations.

This paper describes the Mesosphere Turbulence Experiment (MTeX) that employed two payloads with ionization gauges to obtain four profiles at four different locations within about 30 minutes. The launch condition was a MIL observed by Rayleigh temperature lidar at the launch site. A payload description and first results have been provided by *Collins et al.* (2015). The atmospheric conditions during the launch night, the change in lidar temperatures and sodium densities throughout the night,
which includes a large overturning structure in the sodium layer during the rocket launches are described by *Triplett* (2016). Another publication putting the MTeX results in the larger context of the prevailing stratospheric conditions and an analysis of the gravity wave activity based on lidar observations is forthcoming.

The purpose of this paper is to provide a detailed discussion on the static stability and turbulence structure for each profile following the methods developed for neutral density measurements in the mesosphere and lower thermosphere *Lübken et al.*
(1993); *Rapp et al.* (2001); *Strelnikov et al.* (2003). We also include results from simultaneous chemiluminescent trimethyl aluminum (TMA) releases, including horizontal winds, Richardson numbers, and examples of turbulent structure in the trails (*Larsen*, 2002; *Roberts and Larsen*, 2014).

Our paper is organized as follows. The next section describes the experiment with special emphasis on the derivation of density and temperature profiles. Section 3 presents individual profiles of buoyancy frequency, fluctuations and energy dissipation
rates, as well as wind profiles, wind shears and Richardson numbers. Section 4 discusses our results in the context of other



winter measurements of mesospheric turbulence and multi-scale modeling results. The last section contains our summary and conclusions. An appendix describes results obtained by a small accelerometer for residual drag measurements.

## 2  Experiment

### 2.1  Payloads, salvoes and trajectories

The experiment was designed to study the spatial distribution and temporal evolution of mesospheric turbulence in the presence of a MIL. Two pairs of sounding rockets were launched on January 26, 2015, at 09:13 and 09:14 UT, and 09:46 and 09:47 UT (00:47 LT), respectively, from Poker Flat Research Range, Chatanika, Alaska (65.13 °N, 147.49 °W). The first rocket of each salvo carried the Mesosphere and Lower Thermosphere Experiment (MTeX), an instrumented payload (NASA designation 46.009 and 46.010), while the second rocket of each salvo comprised the Mesospheric Inversion Layer Stratified Turbulence

experiment (MIST) and carried a chemical tracer payload (NASA designation 41.111 and 41.112).

The main instrument on MTeX was the ionization gauge of the Combined sensor for Neutrals and Electrons (CONE) (*Giebeler et al.*, 1993; *Strelnikov et al.*, 2013), which was mounted at the front of the payload together with a suite of plasma instruments on four booms. It was the first time that a CONE sensor was flown on a NASA payload equipped with an attitude control system (ACS). It was also the first time that the same CONE sensor provided upleg and downleg profiles, since the

payload was reoriented near apogee to point the sensor downward back into ram flow. Therefore, the sequence of two MTeX flights provided the first set of four CONE temperature and turbulence measurements obtained in one salvo.

Immediately after nosecone ejection at 52 km and de-spin, the ACS aligned the payload with the velocity vector anticipated for 95 km, halfway in the upleg science window of 70 to 120 km. The spin rate was adjusted close to 2 Hz. The ACS was turned off during the science window in order not to perturb the in situ measurements with cold gas pulses. The ACS was activated

again soon after apogee near 156 km; the payload was flipped over and aligned with the anticipated velocity vector for 95 km on the downleg, halfway in the downleg science window, during which the ACS was turned off again.

The MIST payloads contained TMA canisters for upleg and downleg tracer releases and were only spin-stabilized. The combination of CONE temperature and TMA wind profiles allowed the calculation of Richardson numbers as in the earlier "Turbopause" experiment (*Lehmacher et al.*, 2011).

Figure 1 illustrates the horizontal separation of all four flights. Note the different east-west and north-south scales. The black and red triangles mark the four sets of CONE measurements between 70 and 120 km. Green and orange marks show the location of the tracer releases between 90 and 120 km. For the MTeX flights the altitude of 70 km was reached after 65 seconds on the upleg and 339 seconds on the downleg. The launch azimuth was 0.0° (due north) and 0.6° for 46.009 and 46.010, respectively. The horizontal separation between 70 km upleg and 70 km downleg was about 88 km for 46.009 and 105

30    km for 46.010. As can be seen in the figure, the flights of the second salvo veered slightly westward, but the difference is small in comparison to the extent in the north-south direction.





## 2.2 CONE instrument, neutral densities and fluctuations

The CONE instrument is a spherical hot-cathode ionization gauge designed for pressures up to $\sim 1$ mbar and has been flown over 20 times since the 1990s (*Giebeler et al.*, 1993; *Rapp et al.*, 2001), including on four previous NASA payloads (*Lehmacher et al.*, 2006, 2011). Neutral air is ionized by electron impact and the collected ion current is the primary measurement signal.

Here we include specific details of our methodology, which is similar to the standard procedure (*Rapp et al.*, 2001; *Strelnikov et al.*, 2003). We want to stress the fact, that this is the first application of individual CONE instruments measuring in the ram direction on both upleg and downleg. This will help us in assessing the significance of observed differences in the mesospheric structure.

The ion current varies between $\sim 1$ to 8000 nA, for altitudes from 130 to 65 km, and is measured by a 5-step, auto-ranging
electrometer with 16-bit digitization and 5208 samples per second (or $\sim 0.2$ m at 1000 m/s). The electron emission current from the filament is kept constant at 14 $\mu$A, so that the ion current is roughly proportional to the neutral density. Small deviations from linear behavior were recorded in a calibration vacuum chamber using an MKS Baratron capacitance manometer with $1 \cdot 10^{-6}$ mbar accuracy. We calculated the ion currents using the voltage output for each electrometer range calibrated with a Keithley 261 Picoampere source (G. Krein, pers. communication). We removed a few data spikes due to range switching and
adjusted the voltage offsets in each range to generate a continuous profile for the ion current. Fig. 2 shows the ion currents versus altitude for all four profiles.

The graph clearly shows that the sensitivity of the two CONE ionization gauges is different at lower currents corresponding to altitudes above 100 km, while the variation is very small between 70 and 90 km, indicating similar sensitivity for both gauges. Therefore, we expect similar densities for all four profiles at lower altitudes. During the upleg of 46.010 (orange
profile), we observed strong disturbances of the ion current near 75 and 80 km, and associated disturbances of the emission current (not shown).

The cause of these disturbances is not understood, but simultaneous observations from a small, sensitive 3-axis accelerometer included on the MTeX payload provides indirect evidence for the the existence of a large mesospheric wind. Here we include the main results of this new diagnostic tool. We have deferred the technical details of the accelerometer analysis and comparison
with CONE data to the Appendix.

The accelerometer was mounted close the payload spin axis ($z$) near the center of gravity and observed how the residual drag acceleration decreased with exponentially decreasing density. For flight 41.010, near 75 km, the average $z$ component of the acceleration was 3.7 and 3.3 m$g$ on the upleg and downleg, respectively (Fig. A1). A Direct Monte Carlo Simulation (DSMC) of the supersonic flow using the velocity, density, and temperature conditions for this flight yielded a drag force of $\sim$7.2 N at
75 km, corresponding to a drag acceleration of $\sim$4.0 m$g$, in reasonable agreement with the observations.

The perturbations in the CONE measurement, only on the upleg of 41.010, were unexpected and unprecedented for this instrument. Upon close inspection, we noticed coincident small changes in acceleration of $\sim$0.5 m$g$ near 75 km. No ACS maneuver or other payload event occurred at this time that could have perturbed both measurements, and the angle-of-attack analysis performed by NASA Wallops Flight Facility showed no deviation; therefore, we suggest that a large wind may have





altered the drag force. We performed DSMC simulations adding winds and found that a horizontal wind of 100 m/s, which reduces the ram flow by 30 m/s, can indeed reduce the drag force and the relevant acceleration component by 5% or 0.2 m$g$.

The purpose of the sensor calibration is to correct for the nonlinear variation between pressure and ion current and to account for differences between individual instruments. The calibration for the two CONE ionization gauges is shown in Fig. 3. The

5 gauge used for flight 46.009 was more sensitive above $10^{-2}$ mbar, consistent with the current measurements observed in the flights (Fig. 2). Before applying the calibration information, we reduced the original data rate of 5208 samples per seconds by a factor of 100 and applied a low pass filter to suppress a small modulation with the spin rate of 2 Hz. In order to model the calibration curves, we used a combination of a linear function up to $10^{-2}$ mbar and three Gaussians for higher pressures and converted the currents to pressures and densities. The parameters of the calibration functions were tuned to match a common

density profile below ∼80 km, where atmospheric conditions are most stable over the duration of the experiment.

After applying the calibration, the densities obtained correspond to what is measured inside the CONE ionization volume, and these values are larger than the densities in the free atmosphere due to compression effects in the supersonic flow (*Rapp et al.*, 2001). We apply an aerodynamic "ram" correction that was determined using Direct Simulation Monte Carlo (DSMC) calculations for zero angle-of-attack and altitudes between 120 and 70 km. These "ram factors" vary between 1.6 and 2.6 and

15 were originally calculated for a previous sounding rocket experiment carrying the CONE sensor (*Lehmacher et al.*, 2006). Although the MTeX flights achieved a higher apogee (156 vs. 135 km) and higher Mach numbers ($M \sim 4.5$ vs. 4.0) than the earlier flight, we find that extrapolating these ram factors works well for our flights. Although the ram factors for the CONE sensor are relatively constant at small and moderate angles-of-attack (*Rapp et al.*, 2001), MTeX was the first experiment where the angle-of-attack for CONE was very close to zero due to the use of an attitude control system. Figure 4 shows the densities

after the ram correction, which closely agree with NRLMSISE-00 (hereafter simply MSIS) model densities (*Picone et al.*, 2002). Large wavelike deviations above 100 km, that could already be seen in the current profiles, are significant features in the lower thermosphere.

The temperature profiles $T(z)$ are obtained by integrating the density profile from low to high densities and using the start temperature $T_0(z)$ at 115 km taken from the MSIS model. Only the relative density profile $n(z)/n_0(z)$ matters and the

25 uncertainty in the start temperatures vanishes after 1-2 scale heights (*Rapp et al.*, 2001).

Finally, we calculate the buoyancy frequency as

$$N^2 = \frac{g}{T}\left(\frac{dT}{dz} - \frac{g}{c_p}\right) = \frac{g}{\theta}\frac{d\theta}{dz} \tag{1}$$

Temperature profiles and buoyancy frequencies are discussed in the next section.

Our method to calculate atmospheric densities using calibration data and ram correction follows the standard procedure

(*Rapp et al.*, 2001) and is independent from external data sets (except the start temperature). An alternative method is to use the relative density profile from the nightly averaged Rayleigh lidar signal to normalize the CONE current data, which results in smoother density gradients and temperatures (*Triplett*, 2016).

The open geometry of the CONE ionization gauge aids in the observation of very small neutral density fluctuations ($< 0.1\%$) which are neutral, inert, scalar tracers of turbulence (*Lübken et al.*, 1993). The assumptions and principal methodology of the





spectral analysis has been described by *Lübken et al.* (1993) and relies on observing the transition from inertial to viscous scales in the density fluctuation spectra (*Kolmogorov*, 1941), characterized by the turbulent inner scale $\ell_0$ based on the *Heisenberg* (1948) model.

$$\ell_0 = 9.9 \left(\frac{\nu^3}{\epsilon}\right)^{1/4} \tag{2}$$

While the energy dissipation rate $\epsilon$ (which is determined from $\ell$) can vary over several orders of magnitude, the kinematic viscosity increases exponentially with the scale height. Typical inner scales are 10-50 m, which requires measurements at meter scale resolution to identify the viscous subrange.

As noted already, MTeX was the first experiment with the CONE instrument mounted on an actively stabilized payload, and aligned closely to the velocity vector (angle of attack close to zero). The spin rate was actively reduced to about 2 Hz. It is
well known that small asymmetries in the supersonic flow around the ion gauge lead to modulations of the current signal at the spin frequency and higher harmonics (*Hillert et al.*, 1994; *Strelnikov et al.*, 2003). At 1000 m/s, the payload has moved 500 m during one spin period, which means that the much smaller inner scale is easier to detect at the low spin rate than at common, higher spin rates, e.g., 6 Hz. Also, at larger angles of attack, the spin modulation and higher harmonics cause major interference with the turbulence signal (*Lehmacher et al.*, 2011); therefore, the alignment was important in obtaining good turbulence data.
The time series to be analyzed are relative fluctuations of the ion current $I(t)/I_0$ (identical to relative neutral density fluctuations $n(t)/n_0$ over short intervals), which are determined by subtracting and dividing by a 1000-point (0.2 second) running average. Figure 5 shows as an example the relative fluctuations for the upleg of flight of 46.009.

Two regions of neutral density fluctuations can immediately be recognized around 71 and 76 km altitude. Note that the level of fluctuations is much less than 0.01 (1%). A small spin modulation becomes more prominent above 85 km. The increasing
noise above 90 km was caused by interference from the voltage sweeps of the Langmuir probe on one of the booms (*Collins et al.*, 2015). Turbulent fluctuations, which have larger scale sizes (100s of meters) at higher altitudes, can easily be distinguished from these small scale, regular perturbations.

We have used the wavelet method first applied to CONE data by *Strelnikov et al.* (2003, and references therein), which allows for a finer localization of turbulence layers. The alternative method of calculating Fast Fourier Transform (FFT) spectra
over 1 km or larger intervals can lose some detail in the lower mesosphere but is better at capturing larger scales in the upper mesosphere. An example turbulence spectrum from the lower mesosphere is shown in Fig. 6.

Individual wavelet spectra were averaged over 100-m intervals. The thick black line is such an averaged wavelet spectrum for the interval 71.0 to 71.1 km. The blue line is the least-square fit of a Kolmogorov-Heisenberg spectrum for stationary, homogeneous, isotropic turbulence with slopes $-5/3$ and $-7$ in the inertial and viscous subranges, respectively. Turbulent
spectra were fitted if the data displayed a slope of -5 or steeper in the frequency range between 31.6 to 316 Hz, which is where the viscous subrange can be found. Additionally, some spectra were eliminated if they did not show an inertial subrange with slope -5/3. The red line in the figure indicates the frequency at the fitted inner scale, in this case 21 m. Frequency and scale size are converted via the payload velocity, $f = v/\ell$. Spectra with one standard deviation above and below (dashed lines) were fitted for an error estimate of $f_0$, $l_0$, and $\epsilon$. In this example the lower estimate was 17 m and the upper estimate 27 m. Since



the energy dissipation rate depends on the forth power of inner scale, lower, middle, and upper estimates are 0.68, 1.5, and 3.9 mW/kg, values that are in line with previous measurements of turbulence in winter at high latitudes (*Lübken et al.*, 1993; *Lübken* , 1997).

We will discuss temperature, buoyancy frequency, and turbulence results in Sect. 3.

## 2.3 Chemical tracers

Both MTeX launches were closely followed by two MIST payloads for wind and turbulence measurements in the lower thermosphere. TMA trails were released on the upleg and downleg between ∼80 and 150 km. Cameras for ground-based photography of the trail were located at Poker Flat, Coldfoot (67.25 °N, 150.15 °W), and Toolik Lake (68.63 °N, 149.60 °W). For a review of the technique see *Larsen* (2002). Typical errors of horizontal wind components are 5 – 10 m/s. In the next section, we show wind profiles calculated from the upleg trails and examples of trail structures as they relate to the observed winds and temperatures.

## 3 Results

### 3.1 Temperatures

Figure 7 shows all four temperature profiles derived from CONE densities combined in one plot. The start temperature is chosen from the MSIS profile at 115 km; given the large variations around 105 km, we estimate an uncertainty of 30 K at 115 km, which is larger than the instrumental error, and decreases exponentially towards lower altitudes (*Rapp et al.*, 2001). The four profiles are similar, which is expected given the moderate horizontal and temporal separation (see Fig. 1). (For the upleg of 46.010 (orange), we interpolated the densities logarithmically in the two regions, where the CONE currents were disturbed. The gaps are shown with dashed lines.)

Characterizing the profile with large brush strokes, we observe a relatively warm winter mesosphere up to 80 km, a quasi-adiabatic region between 80 and 88 km, another stable region up to 95 km, followed by a second quasi-adiabatic region up to the mesopause at  170 K and  102 km. The two bottommost regions agree well with the Rayleigh lidar temperatures (*Triplett*, 2016). The lower thermosphere is unusually structured; the upleg profiles have temperature excursions up to 60 K warmer than MSIS between 105 and 110 km. The mesopause is markedly colder than MSIS; a feature that we also observed during an earlier winter experiment together with significant large scale, long period wave activity (*Lehmacher et al.*, 2011). In the graph, we included two SABER temperature profiles with tangent points closest to the rocket observations, but obtained about 2 and 3 hours earlier. SABER stands for Sounding of the Atmosphere using Broadband Emission Radiometry and is an instrument on the Thermosphere Ionosphere Mesosphere Energetics Dynamics satellite (TIMED) in operation since 2002. SABER data were retrieved from the data server at saber.gats-inc.com. Despite a very different technique and sampling geometry for a satellite limb sounder, both SABER profiles show remarkably similar structures: a distinct and relatively cold mesopause, and two quasi-adiabatic regions bracketing a stable region.





Figure 8 shows details of the CONE temperature profiles. For this plot, the in situ data have not been low-pass filtered; therefore, flight 46.009 shows some spin modulation that could also be seen in Fig. 5. On the other hand, temperature fluctuations below 80 km are a clear indication of the sub-kilometer dynamics in the mesosphere that only in situ instruments can detect. The 46.009 downleg profile (blue) differs significantly from the other profiles near 72 km, while the upleg profiles, which
were closest together, agree well (if we ignore the interpolated altitude intervals). Consecutive lidar profiles showed that the wave activity, based on the lidar temperatures during the night, was relatively weak and at the time of the launches only small inversion layers near 61 and 70 km were present over the launch site. The nightly lidar average also reproduces the strongly negative temperature gradient above 80 km (*Triplett*, 2016).

### 3.2  Buoyancy frequency and turbulence

Figures 9 – 12 compare the temperature and turbulence structure side by side for each profile. The first panel in each figure shows the temperature profiles based on two types of analysis. The thick profile is identical to the one shown in Fig. 7. The derivation takes into account the calibration and ram correction as described above. The thin temperature profile is derived using the nightly average lidar profile. The difference between the profiles is shaded and may serve as an estimate of the total absolute temperature error, which is between 5 and 10 K. (The uncertainty due to the start temperature is negligibly small
below 95 km.) The green line is an MSIS model profile. The second panel shows the square of the buoyancy frequency $N^2$ derived from the temperature profiles. It is important to note that the $N^2$ profile is quite robust since the location of stable and unstable regions is independent of the absolute temperature. The red line at $N^2 = 0$, the adiabatic temperature gradient, serves as a guide for instability, the outer green line at $N^2 = 8 \times 10^{-4} \mathrm{s}^{-2}$ indicates very stable conditions. The third panel shows the spectrogram of the global wavelet spectra of the neutral density fluctuations at 100 m resolution, as described above. Above
85 km, signal modulations with the spin frequency and harmonics at 2, 4, and 6 Hz become significant as can already be seen in Fig. 5 and Fig. 6. A white line near the bottom indicates the "cone of influence", where wavelet power cannot be estimated (*Torrence and Compo*, 1998). The fourth panel shows the energy dissipation rates $\epsilon$ derived for 100 m intervals with upper and lower estimates, as explained in Sect. 2.

The first profile was obtained on the upleg of flight of 46.009. Two distinct layers of turbulence were observed centered
around 71 and 75 km, respectively. Energy dissipation rates ranged from 0.18 to 6.4 mW/kg and the median values in the lower and upper layer were 1.5 and 2.7 mW/kg, respectively. The lower layer is mostly above a small local temperature maximum at 71 km associated with an inversion layer. The upper layer is near a smaller local temperature maximum. A second inversion layer was observed near 80 km with a temperature maximum near 81.5 km. There are density fluctuations between 83 and 84 km in a region of low stability, but the spectrum did not the fulfill the criteria for determining an inner scale and turbulent
energy dissipation rate.

The second profile was observed on the downleg of flight of 46.009, 70 km north of the upleg profile. The temperature profile is significantly different in the lower mesosphere and has a broad maximum at 72.5 km associated with a deep turbulence enhanced layer. The maximum energy dissipation rate is found at 72.8 km with 36 mW/kg, slightly above the temperature maximum. The median value for the entire layer is 2.2 mW/kg. It appears that the two regions of turbulence generation observed





on the upleg are merged at this location; however, this cannot be verified without additional observations at intermediate locations. The inversion layer at higher altitudes is very distinct in this profile with a clear maximum of the buoyancy frequency near 80 km, slightly lower that on the upleg. Again, the fluctuation spectra show no evidence for turbulence in the upper mesosphere.

The third profile was obtained on the upleg of flight of 46.010. Its location was very close to the first profile and the measurement occurred 33 minutes later. As explained above, we do not have a complete density and temperature profile due to the anomaly of the CONE sensor, and the interpolated regions are marked with dashed lines. However, it can be assumed that the temperature profile did not change dramatically compared to the first profile, as can be seen below 73 km and also above 82 km where the measurements were undisturbed. This is also confirmed by the series of Rayleigh lidar profiles. With this

caveat in mind, we did the wavelet analysis of the ion current fluctuations in the perturbed regions and found that the spectra conformed with our turbulence model, although the fluctuations were much amplified in the perturbed regions, as can be seen from the red contours in the spectral plot. Compared to the 46.009 upleg, the turbulent layer near 70 km had weakened or moved by advection, although there is still a thin layer present at 70.5 km with 1.8 mW/kg. These data were not affected by the anomaly. The layer around 75 km from the 46.009 upleg may have expanded; it stretches now from 74 and 77 km. The

dissipation rates are slightly smaller and reach only 2.0 mW/kg. The biggest difference in the 46.009 upleg is observed near 80 to 81 km, where no turbulence was observed 33 minutes earlier. On the 46.010 upleg, this region exhibits strong fluctuations with inner scales corresponding to energy dissipation rates up to 1.3 mW/kg. The temperatures in the undisturbed region near 82 km suggest superadiabatic conditions. Further support for strong mixing in this region comes from the sodium densities and mixing ratios observed by lidar (*Triplett*, 2016). Low sodium density air was mixed upward in a major overturning event and

extended from 81 to 88 km during the two flights. This agrees strikingly well with the quasi-adiabatic region observed in all four in situ profiles.

The last profile was obtained on the 46.010 downleg, about 10 km west of the first downleg. The temperature profile agrees in many details with the other three profiles, most significantly, the local maximum near 80 km below a deep layer with low static stability, as just discussed. At the lower altitudes we find very weak turbulence in several narrow layers near 70 km, 72.5 km,

and 74 km, mostly associated with stable regions of the atmosphere. The largest value is 2.3 mW/kg at 74.2 km in a very thin layer. The region of turbulence in the lower mesosphere is broadly consistent with the observation from the 46.009 downleg, but the intensity level is weaker. In contrast to the earlier downleg profile, two strong, but narrow layers were observed near 80 and 84 km with maximum epsilon values of 4.0 and 5.6 mW/kg. The lower of these layers coincides with the local temperature maximum near 80 km.

In summary, and perhaps not surprisingly, we observe strong similarities in the large scale temperature and stratification structure, but great variability in the altitude, thickness, and strength of the fluctuation layers. Turbulent spectra are found in the more stratified region below 80 km, while fluctuations and turbulence are largely absent in the well-mixed layer above.



### 3.3 Neutral winds, Richardson number, and trail structure

Figure 13 shows the zonal, meridional and total horizontal wind profiles obtained during the upleg releases as red, blue and black lines, respectively. Common features are an extreme westward zonal wind shear near 110 km and strong westward winds above. Below 105 km, winds were smaller and relatively constant. Above 110 km, the winds significantly changed between the first and second flight; the zonal component weakened and the meridional component shifted southward. The flights occurred under moderately active conditions and a bright auroral arc. High southwestward wind speeds of 200 m/s above an extreme zonal wind shear were also observed during the ARIA II experiment under moderate to high geomagnetic activity (*Larsen et al.*, 1997).

Another presentation of the wind components are the hodographs in Fig. 14. Symbols mark altitudes in 1 km steps; the big, filled circles mark 90, 100, 110, 120, and 130 km. The lowest altitudes start on the right and the wind vector rotates clockwise with increasing altitude. Particularly, MIST-2 showed a consistent rotation up to 110 km, typical of a tidal or inertia-gravity wave, but stretched out by the strong westward shear. A similar wave without any additional shear was observed during geomagnetically very quiet conditions (*Lehmacher et al.*, 2011).

The simultaneous measurements of temperatures with the CONE instrument and winds with the chemical tracer technique allow the calculation of Richardson numbers as an index for instability in the turbopause region (*Lehmacher et al.*, 2011). This is only the second experiment for which this combination of measurements was available. We interpolated CONE upleg temperatures in 1-km intervals to match the upleg wind data and calculated the Richardson number as

$$Ri = \frac{N^2}{(du/dz)^2 + (dv/dz)^2} \tag{3}$$

In Fig. 15 and 16 we show profiles of buoyancy frequency, horizontal wind shear, and Richardson number at altitudes, where we have simultaneous temperature and wind data. For the first salvo, we find a minimum in the Richardson number of less than 0.1 at 110 km. At this altitude, the buoyancy frequency was unusually low, paired with an extreme wind shear. For the second salvo, similar conditions existed at 107 km. All four temperature profiles (Fig. 7) showed regions of warmer temperatures in the lower thermosphere between 102 and 110 km, most prominently in the upleg profiles.

The extreme wind shear is also directly visible in the images of the puffed TMA trails. Figure 17 shows the upleg trail viewed from the North from Toolik Lake (left) and from below from Poker Flat (right). About 70 seconds after the release, the chemiluminescent material at 110 km was already stretched out (shown by the red arrows). In this region, above the turbopause, no small scale irregularities were visible. The Reynolds number is small in this region and the flow remains laminar (*Blamont and de Jager*, 1961).

Below ∼103 km, the trails often develop billows of large and small sizes due to atmospheric turbulence (e.g., *Blamont and de Jager*, 1961; *Roberts and Larsen*, 2014). A very clear example can be seen in both downleg trails as viewed from Toolik Lake (Fig. 18). These images were taken 190 seconds after TMA was released at 100 km on the downleg (red arrows). The trails between 95 and 100 km appear as vertical billowing columns with a defined top. The temperature structure measured 50 km further south shows a quasi-adiabatic lapse rate between 95 and 100 km and a very stable layer above (see Fig. 7). Considering both TMA images and temperature structure, this suggests the presence of a deep convective layer just below the mesopause.



*Roberts and Larsen* (2014) used the entire downleg trail below the turbopause to determine the structure function coefficient as a function of scale size, while the large scale temperature structure was unknown. Our case presents an opportunity to study the evolution of the structure function under known stability conditions.

## 4 Discussion

The MTeX experiment was the first time that four profiles of in situ neutral turbulence and background temperature were obtained close together in time and space. While this is still a very small sample of the turbulent flow field, it allows a limited comparison with high-resolution multi-scale gravity wave breaking simulations.

*Fritts et. al.* (2018a, b) presented cases of the interaction between a small-scale monochromatic gravity wave ($\lambda_x = 20$km, $\lambda_z = 20$ or $40$km) with a MIL, which was centered at 80 km and spanning about 20 km.

The numerical simulations showed that the interaction was relatively weak and did not create several kilometer deep layers of instability or weakened stability. However, it was also found that the impact of turbulent heat fluxes depends on the generation mechanism; gravity wave breaking occurring in the low stability phase had less impact than KH instability-generated turbulence.

During our experiment, we did not encounter a large MIL of the type set as the temperature background in the simulation.
However, the mesosphere was on average stable below 80 km and very weakly stable or quasi-adiabatic between 80 and 88 km. Almost all of our turbulence layers were observed in the more stable region below 80 km. The perturbations in stability due to GW interactions resemble the individual $N^2(z)$ profiles shown by *Fritts et. al.* (2018b) in their Fig. 13. Our upleg and downleg results show little relation; patchiness is expected for turbulence over this spatial domain, and is also found in the multi-scale simulations. We also found significant differences in the turbulence strength and layer distribution between the first and second
flight. Numerical simulations of gravity wave breaking show significant evolution over 30 minutes (∼5 buoyancy periods).

Comparing values for the energy dissipation rate, the statistics of winter turbulence measurements obtained at Andøya (*Lübken et al.*, 1993; *Lübken* , 1997) was recently extended and updated by *Szewczyk* (2015). Average energy dissipation rates increase continuously from 1 mW/kg at 70 km to 10 mW/kg at 80 km, up to a maximum of 50 mW/kg at 90 km, while the total variability envelopes cover almost 4 orders of magnitude. Our values fall well within this range; however, the absence of
25 significant turbulence above 80 km seems unusual during our flights. As mentioned earlier, *Triplett* (2016) found that gravity wave activity in the 40-50 km region during this period was extremely low and suggested that this may have contributed to reduced gravity wave breaking and turbulence activity in the upper mesosphere.

In an earlier experiment from Poker Flat, fluctuation activity was small in the lower mesosphere, despite a prominent mesospheric inversion layer at 70 to 75 km (*Lehmacher et al.*, 2011; *Collins et al.*, 2011). An overturning event in the sodium layer
coincident with a near adiabatic layer between 75 and 80 km suggested that it may have been accompanied by strong downward turbulent heat flux (*Collins et al.*, 2011), however, the new simulations by *Fritts et. al.* (2018b) did not produce significant heat fluxes above the temperature inversion. On the other hand, a turbulent layer was observed in this earlier experiment between 88 and 90 km, with energy dissipation rates up to 30 mW/kg, observed in neutral and electron fluctuations.





Another winter case study was presented by *Szewczyk et al.* (2013). A strong temperature inversion between 86 and 89 km and quasi-adiabatic layer between 89 and 91 km was strongly turbulent, especially in the adiabatic region. It was concluded that gravity wave breaking and turbulent heating was creating or maintaining the inversion layer, also at odds with the recent modeling results by *Fritts et. al.* (2018b). Previously, *Liu et al.* (2000) had carried out a 2-D modeling study of gravity wave-

5 tidal interaction that produced extremely high GW heat fluxes and adiabatic gradients. The model was set up to test the hypothesis that this type of wave-wave interaction can generate and maintain MILs. The model results suggested that direct turbulent energy dissipation was small compared to the overall heating rates obtained in the model.

*Lehmacher and Lübken* (1995) reported the results of a mid-latitude study of turbulence generation in a deep, partially super-adiabatic layer between 75 and 80 km. This suggests that small-scale turbulence can be important in initial gravity wave

breaking and mixing in the mesosphere, but once the layer is well-mixed, turbulent fluctuations are largely absent. It is true that for strictly vertical adiabatic motions on top of an adiabatic background, i.e., $N^2 = 0$, density fluctuations cannot be observed, since

$$\frac{\delta n}{n} = \frac{N^2}{g} \delta z \tag{4}$$

*Szewczyk* (2015) presented a statistical study of a large number of in situ neutral turbulence profiles, which showed that

mesospheric turbulent density fluctuations have a normal distribution over buoyancy frequency with a maximum near $dT/dz \sim -5$ K/km. However, there is still a significant number of cases, where $dT/dz$ is between $-9$ and $-11$ K/km.

A recent sounding rocket flight from Andøya was equipped with two CONE instruments to provide measurements on the upleg and downleg (*Strelnikov et al.*, 2017). This was in summer, however, when the temperature and turbulence structure is different than in winter (*Lübken et al.*, 1993). In this case, a relatively large variability in temperatures was observed between

20 the upleg and downleg portions of the flight, which were separated by 41 km at 70 km altitude. Other observations, such as VHF radar echoes and winds, also showed large horizontal variability. It was suggested that a gravity wave with 30-km horizontal wavelength could have modulated the temperature field and associated turbulence generation.

In both mesospheric turbulence experiments from Alaska (2009 and 2015) we have observed nearly adiabatic layers in the upper mesosphere accompanied by overturning events at the bottom or middle of the sodium layer, respectively. Such

structures in the sodium layer have been modeled and are thought to be associated with large-scale gravity waves, that are not or not breaking, either partially or fully (*Xu et al.*, 2006). The occurrence and strength of turbulence in gravity wave breaking has also been extensively modeled by *Achatz* (2007). It was found that gravity waves may generate weak turbulence even before becoming statically or dynamically unstable. Another interesting result in the study by *Achatz* (2007) is that the statically enhanced roll mechanism plays an important role in the energy exchange for the breaking of inertia-gravity waves,

which are often observed in the winter upper mesosphere (*Meyer et al.*, 1987).

A large number of winter measurements confirms that strong isotropic turbulence is rarely observed above 95 km. *Szewczyk* (2015) shows that between 90 and 100 km small scale turbulence is only observed with 16 % probability in high-latitude winter, however, energy dissipation rates are most likely between 10 and 100 mW/kg. Therefore, it is surprising that during



MTeX we find visual evidence for deep convection between 95 and 102 km in the TMA trails. A closer examination of the structure function derived from the TMA images (*Roberts and Larsen*, 2014) is needed to shed further light on the nature of the turbulence near the turbopause.

## 5   Summary and conclusions

MTeX was the first sounding rocket experiment that obtained four in situ temperature and neutral turbulence profiles within 33 minutes in the winter mesosphere. The four temperature profiles showed a high degree of consistency at large scales. Two relatively stable regions existed between 68 and 82 km and between 88 to 95 km and two nearly unstable regions between 82 and 88 km and again between 95 to 102 km. The temperature structure was also observed by Rayleigh lidar up to 90 km (*Triplett*, 2016). In the nearly-adiabatic region between 82 and 88 km, neutral sodium was well mixed in a large-scale

overturning event (*Triplett*, 2016), which could have been be associated with a large-scale gravity wave that was not fully breaking (*Xu et al.*, 2006).

Between 85 and 115 km, we obtained simultaneous wind measurements from TMA tracer trails and were able to derive Richardson numbers as a measure of dynamical instability. This was the second experiment in which we obtained Richardson numbers from the combination of ionization gauge temperatures and TMA winds. While the earlier "Turbopause" experiment

was conducted under geomagnetically quiet conditions, but during significant gravity wave activity (*Lehmacher et al.*, 2011), the MTeX and MIST flights had moderately active conditions in the presence of a bright auroral arc. Temperatures above the mesopause were highly disturbed, extreme easterly wind shears were observed at 108 to 110 km, and easterly winds of 200 m/s persisted above 110 km, as in the earlier ARIA II experiment (*Larsen et al.*, 1997).

The stable region between 68 and 82 km did not have a persistent positive temperature gradient as in major MIL events and

as modeled by *Liu et al.* (2000) and *Fritts et. al.* (2018b). However, most turbulent layers were found in this region and there was almost no turbulent activity in the weakly stable region above, in agreement with the modeling in *Fritts et. al.* (2018b). The turbulent energy dissipation rate was 1–10 mW/kg, in agreement with many previous in situ neutral turbulence measurements in the winter mesosphere (*Lübken* , 1997; *Szewczyk*, 2015)

The experiment confirmed that the winter mesosphere is highly variable, and on the day of the experiment, gravity wave

activity in the upper stratosphere and lower mesosphere was lower than normal (*Triplett*, 2016). During our launches, a persistent MIL was not present between 70 and 80 km, where they are often observed (*Meriwether and Gerrard*, 2004). (However, the SABER temperature profiles shown in Fig. 7 suggest that an extended MIL was present between 90 and 100 km, possibly of the tidally induced, "upper" MIL type. In order to gain a better understanding of the relationship between turbulent energy dissipation and other quantities relevant for describing turbulent activity, such as heat flux and Prandtl numbers, more multi-

point observations of turbulence are needed. Temperature measurements in the mesosphere should be accompanied by wind measurements with similar resolution in order to derive detailed gravity wave parameters and Richardson numbers. It is also possible to construct additional modeling cases based on our observations.



*Data availability.* The sounding rocket experiment was funded by NASA's Heliophysics program. In accordance with NASA's data sharing policy, the data sets are public. The data that support the findings of this study are available from the corresponding author on reasonable request. SABER temperature profiles are available at http://saber.gats-usa.net/. MSIS profiles were obtained at https://ccmc.gsfc.nasa.gov/modelweb/models/

## Appendix A: Accelerometer data analysis

A small 3-axis MEMS accelerometer (Type Kionix KXR94-2050) with a sensitivity 1 $g$/V ($g = 9.81$ ms$^{-2}$) and range $\pm 2.5$ $g$ in each channel was included in the payload and mounted on the longitudinal payload axis and close to the center of gravity. The voltage output was amplified for a maximum range of 500 m$g$ and a nominal bit resolution of 0.015 m$g$. We estimated that the sensitivity was sufficient to detect the variation in drag acceleration below altitudes of 80 km and included the accelerometer as proof-of-concept experiment. The signal from all three axes was sampled at 5208 Hz. The typical noise density (according

to the manufacturer's data) is 0.045 m$g$Hz$^{-1/2}$.

Figure A1 shows the accelerometer data (raw data in grey and smoothed data in cyan) compared with the CONE ion currents (red) for flight 46.010. First, the CONE currents are almost identical except for the major perturbations around 75 and 80 km discussed in the main text and Fig. 2. The good agreement between all four current profiles suggests that atmospheric densities are not very different for upleg and downleg in the lower mesosphere.

Next, we show the acceleration component along the z-axis (payload spin axis). Please note that we have subtracted a constant bias from the accelerometer data, which was determined at altitudes above 120 km from the raw data. For both upleg and downleg, the acceleration residuals decrease exponentially with altitude. The upleg portion of flight 46.010 shows a small spin modulation (grey line), which is consistent with a larger coning half angle of $1.3°$ observed by the onboard gyroscope (as compared to $0.5°$ on downleg). On the other hand, the stable attitude on the downleg begins to deteriorate at 70 km, which is

visible in the beginning spin modulation in the downleg accelerometer data. The cyan line is a running mean to reduce this spin modulation and noise. Considering these flight conditions, we suggest that the smoothed, residual accelerations can be interpreted as measure of the atmospheric drag force on the payload, which we write as

$$\mathbf{F} = m\mathbf{a} = \frac{\rho C_d A}{2}|v|\mathbf{v} \tag{A1}$$

where $m$ is the payload mass, $\rho$ the atmospheric density, $C_d$ the drag coefficient, $A$ the cross sectional area of the payload,

and $\mathbf{v}$ the payload velocity. Since the payload velocity was aligned with the accelerometer $z$ axis, almost all drag was registered in the $z$-channel of the 3-axis accelerometer. The $x$ and $y$ channels registered less than 1 milli-$g$ throughout these stable portions of the flight.

A Direct Monte Carlo Simulation (DSMC) of the supersonic flow using the velocity, density, and temperature conditions for this flight yielded a drag force of 7.2 N at 75 km. We used the NASA DAC97 package for our simulations (*LeBeau*,

1999). Dividing the force by the payload mass of 187 kg, this corresponds to an acceleration of 4.0 m$g$, which is close to the observed (average) accelerations of 3.7 and 3.3 m$g$ on upleg and downleg. This calculation and the exponential decrease of the acceleration values gives us confidence that the accelerometer signal is due to drag acceleration.



The perturbations in the CONE measurement were unexpected and unprecedented, and the simultaneous change in acceleration of 0.5 m$g$ (especially near 75 km) can provide additional clues. No ACS maneuver or other payload event occurred at this time that could have perturbed the measurement, therefore it is suggested that a large wind may have altered the drag force. We performed DSMC simulations adding winds and found that a horizontal wind of 100 m/s, which reduces the ram

flow by 30 m/s, can indeed reduce the drag force and the relevant acceleration component by 5% or 0.2 m$g$. Additional vertical winds could add to this change. Qualitatively, it seems plausible that a strong wind could have caused a small change in the drag force, and possibly also the disturbance in the CONE ionization gauge, which is directly exposed to the flow.

While sensitive accelerometers on supersonic free-falling spheres have been used previously to successfully measure winds, densities and temperatures in the mesosphere and lower thermosphere (*Philbrick et al.*, 1985), this experiment demonstrates

that changes in atmospheric drag may be observed for much heavier, cylindrical payloads with a low-cost device, however, only in the denser mesosphere. A similar accelerometer experiment was flown on the German student mission MAPHEUS-1 (*Stamminger et al.*, 2009), which appears less sensitive than our device. More sensitive and lower-noise accelerometers could provide a basic method for routine wind and density measurements in the mesosphere.

*Competing interests.* The authors declare that they have no conflict of interest.

*Acknowledgements.* This research was supported by NASA grants NNX13AE35G (Embry-Riddle Aeronautical University), NNX13AE26G and NNX14AH45G (Clemson University) and NNX13AE31G (University of Alaska Fairbanks). The CONE sensors were built by H.-J. Heckl and calibrated by A. Szewczyk at the Institute for Atmospheric Physics. The CONE electronics was designed and built by von Hoerner & Sulger GmbH, Schwetzingen. We thank NASA Wallops Flight Facility and Poker Flat Research Range for mission and payload design and launch and recovery operations.



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





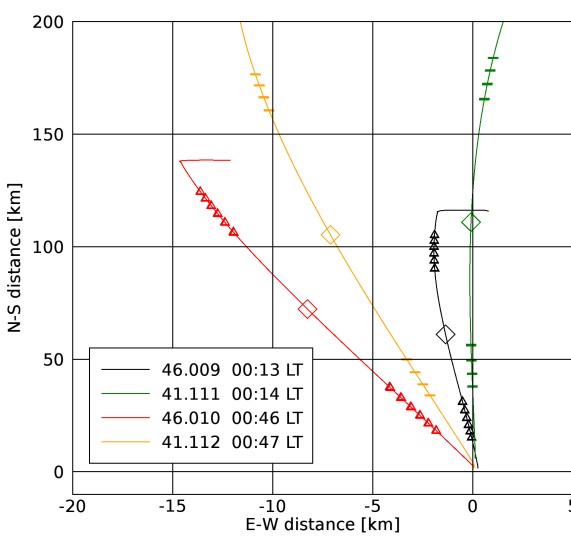

**Figure 1.** Horizontal projection of the four sounding rocket trajectories. The triangles mark altitudes 70 to 120 km in 10-km steps on upleg and downleg for the instrumented payloads (MTeX 46.009 and 46.010), and the dashes mark altitudes 90 to 120 km for the chemical tracer payloads (MIST 41.111 and 41.112). The diamonds mark the apogees.





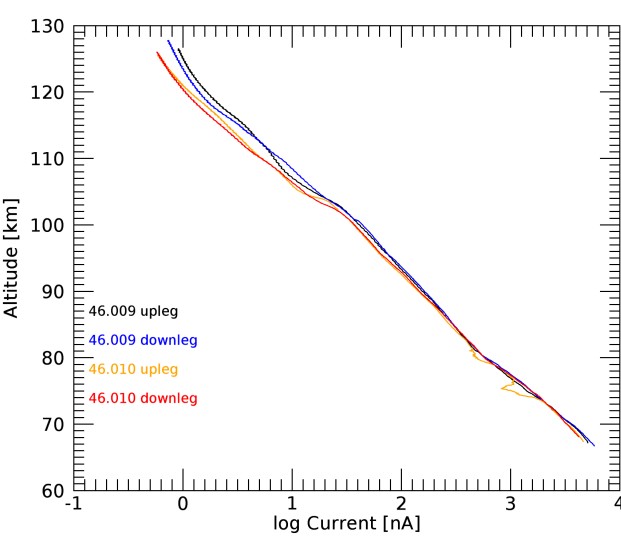

**Figure 2.** Four profiles of ion currents observed during the MTeX flights. Each pair of upleg and downleg profiles were obtained with a single CONE instrument.





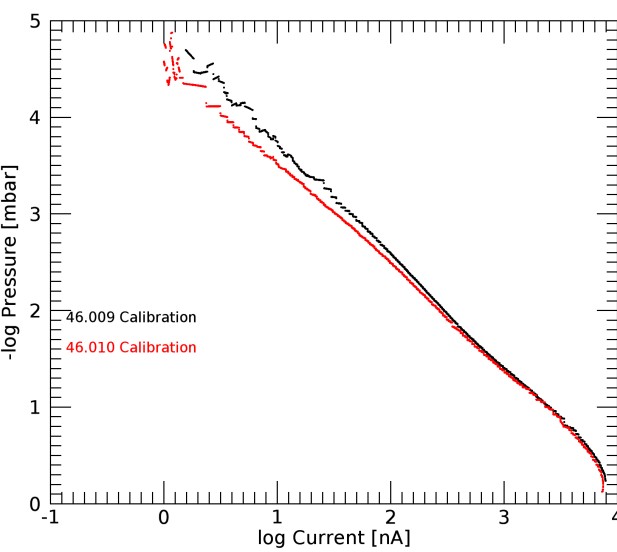

**Figure 3.** Ion currents observed during laboratory calibrations of the two CONE sensors. Irregularities at pressures below $10^{-3}$ mbar in one of the profiles are due to irregularities in the gas flow into the chamber and can be ignored.





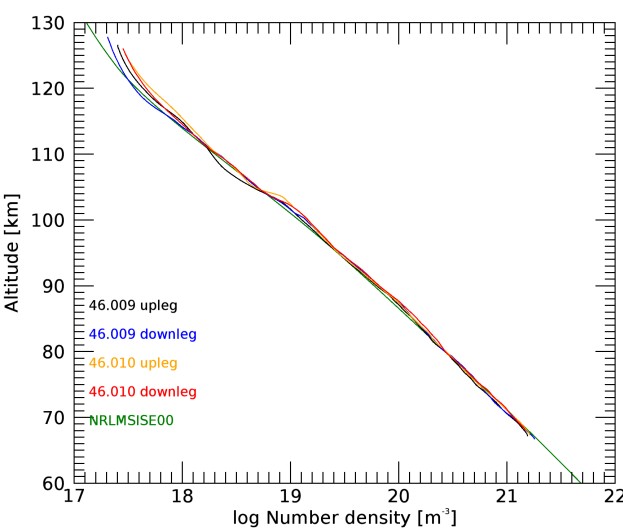

**Figure 4.** Neutral number densities for four profiles after calibration and ram correction. The four profiles agree now for all altitudes, but display significant variability above 100 km.





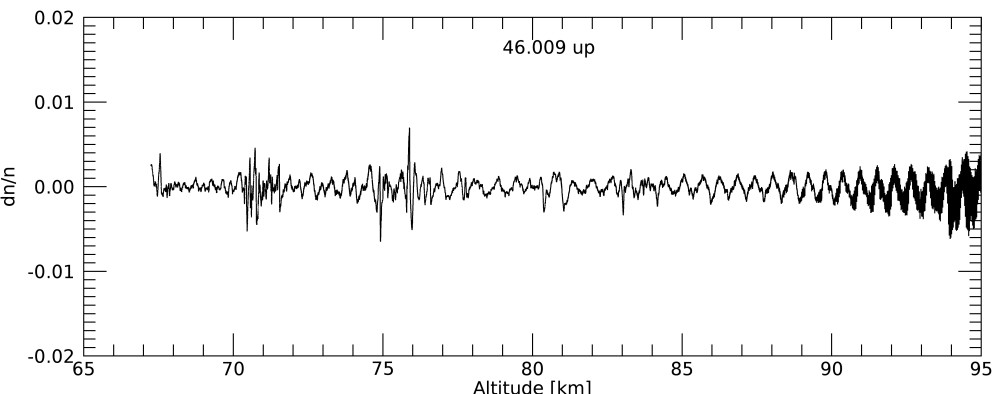

**Figure 5.** Example of CONE relative density fluctuations for flight 46.009 upleg.





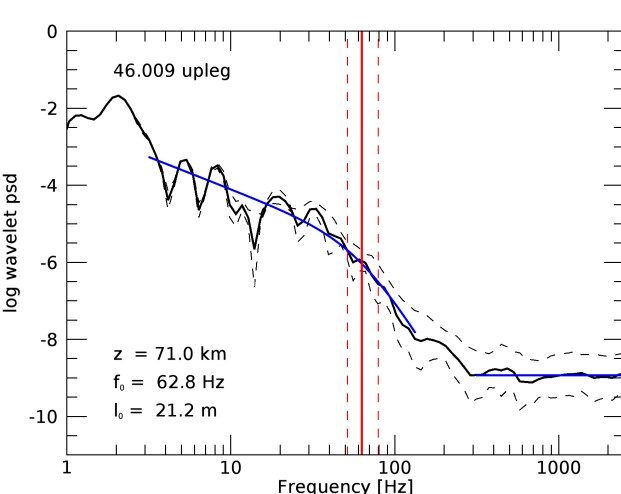

**Figure 6.** Example of wavelet spectrum of turbulent fluctuations. The blue curve is a best fit of a theoretical spectrum including the transition from the inertial subrange to the viscous subrange. For details see text.





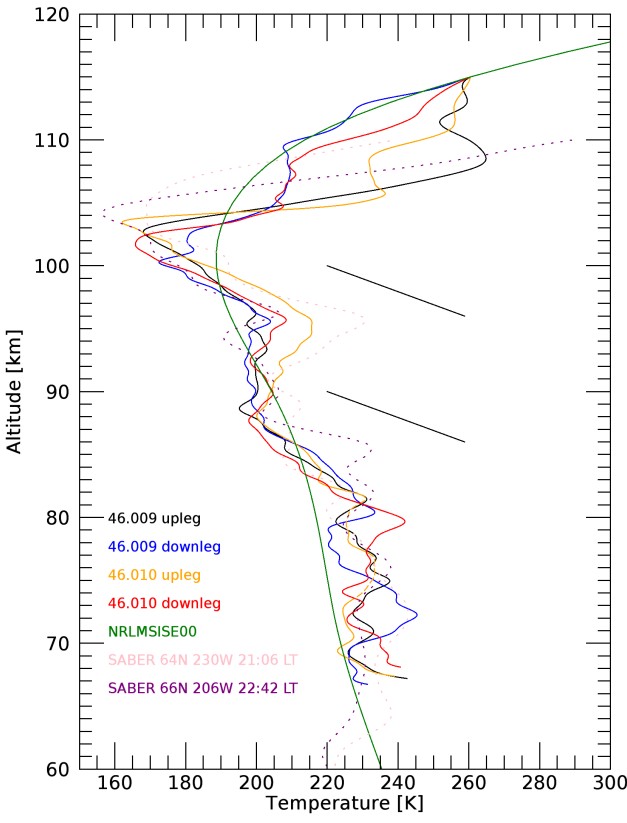

**Figure 7.** Temperature profiles (solid lines) derived from densities in Fig. 4. Start values at 115 km were chosen from MSIS (green line). Individual SABER temperature profiles (dotted lines) obtained during that night in this area show good agreement with the general temperature structure. The legend lists times and tangent point location for these profiles. The two straight solid lines indicate the adiabatic gradient of −9.7 K/km.





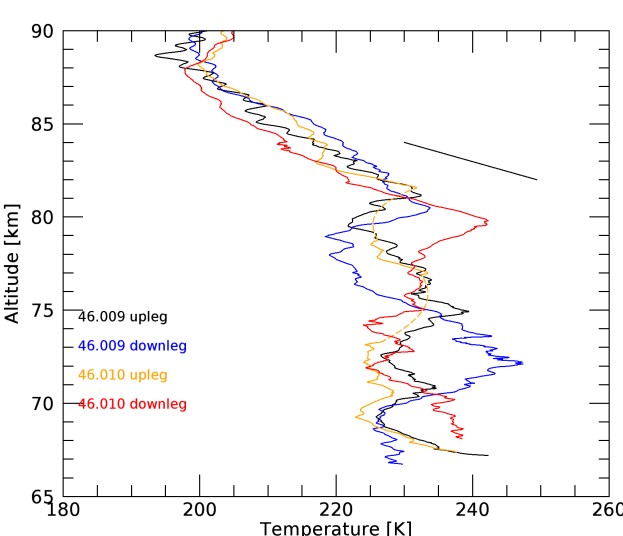

**Figure 8.** Detail of temperature profiles in the lower mesosphere. Same data as in Fig. 7, but unfiltered to emphasize fine structure. The regular modulations above 80 km in the black profile are due to the payload spin.



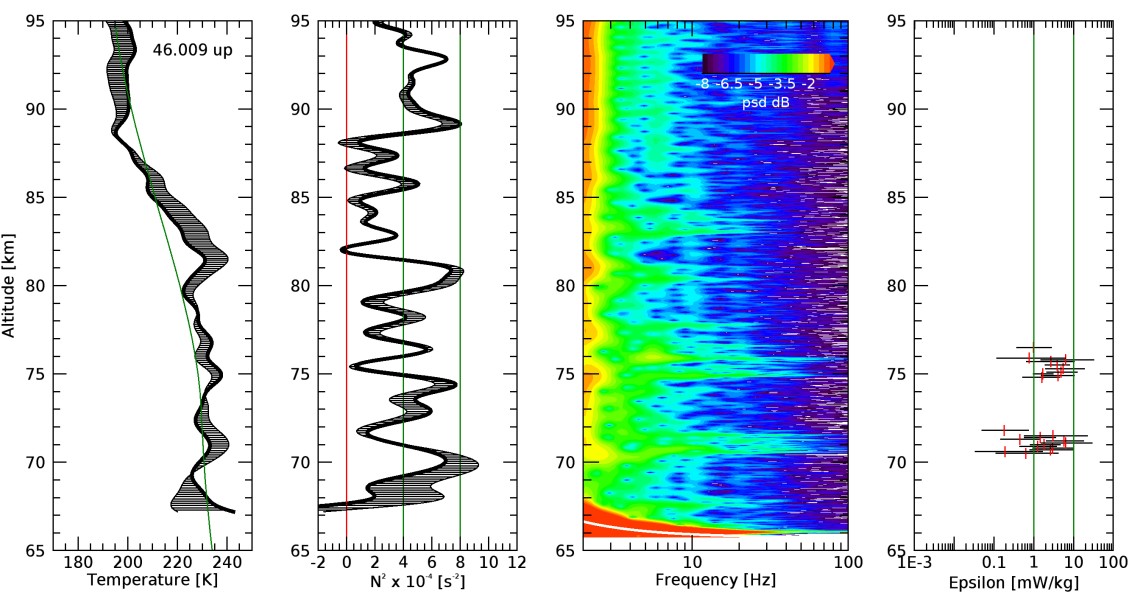

**Figure 9.** Temperature profile, buoyancy frequency (both with bands of uncertainty), wavelet spectra of neutral density fluctuations, and turbulent energy dissipation rates for flight 46.009 upleg. For details see text.





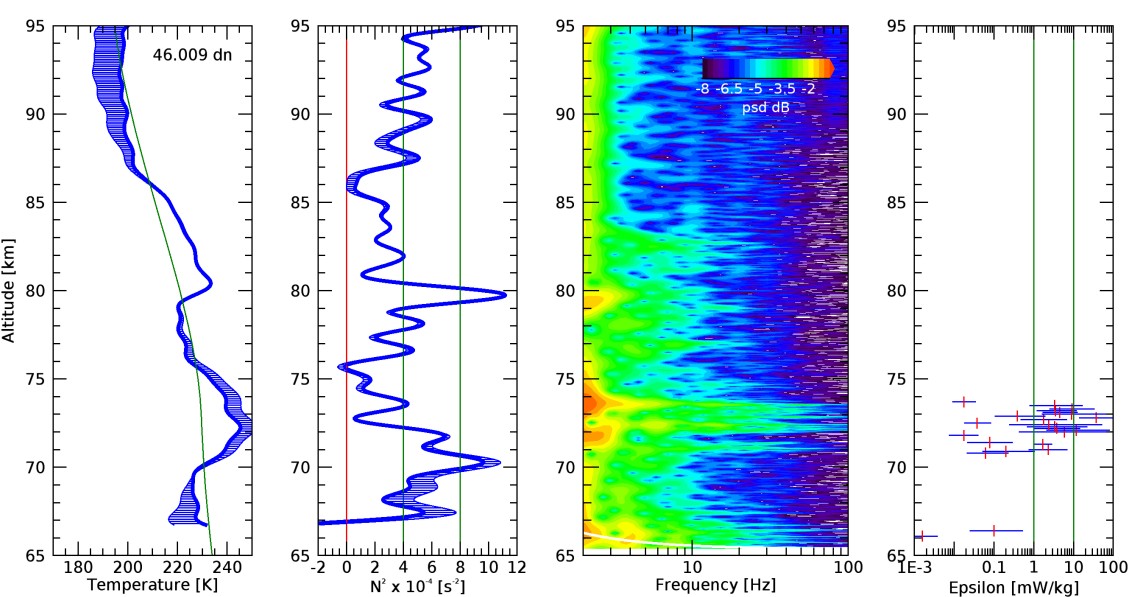

**Figure 10.** Same as Figure 9, but for 46.009 downleg.





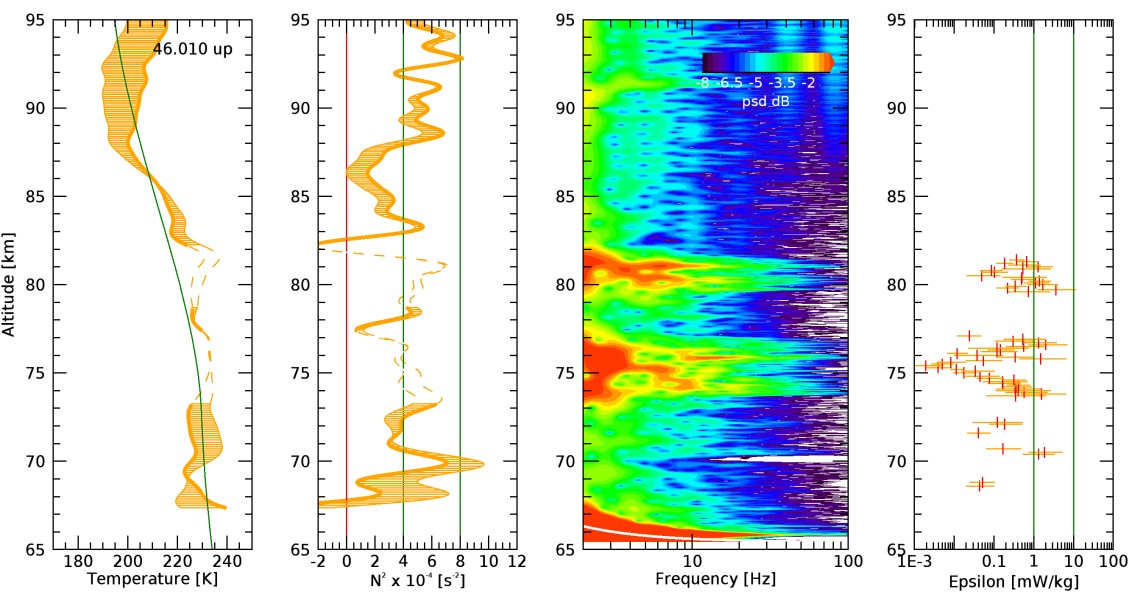

**Figure 11.** Same as Figure 9, but for 46.010 upleg.

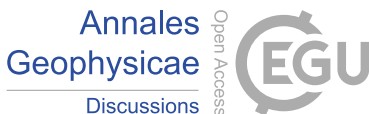

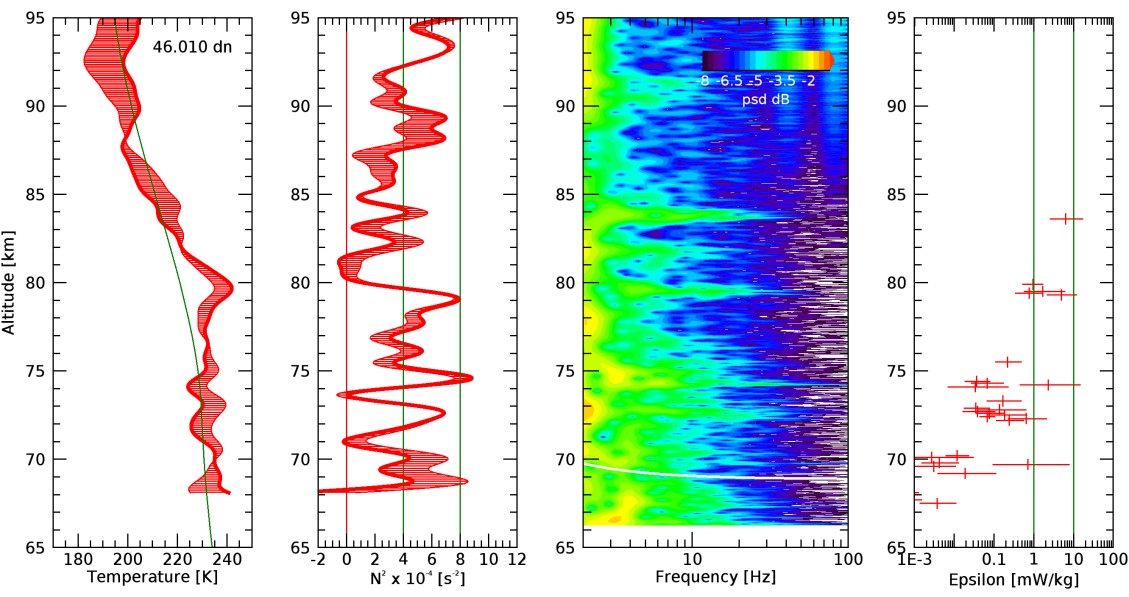

**Figure 12.** Same as Figure 9, but for 46.010 downleg.





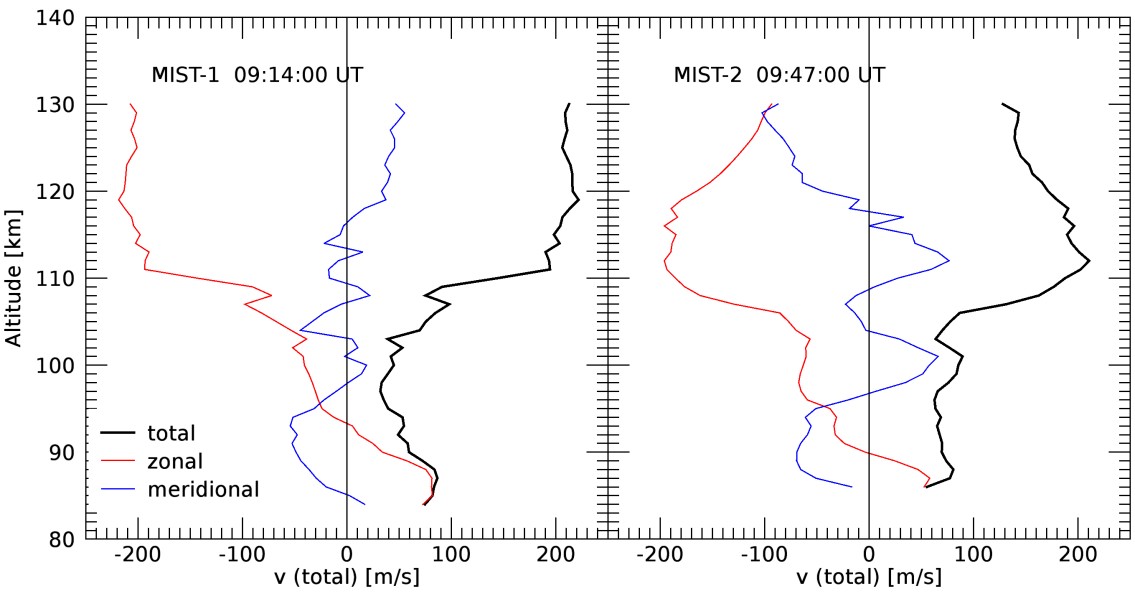

**Figure 13.** Horizontal winds derived from chemical trails.





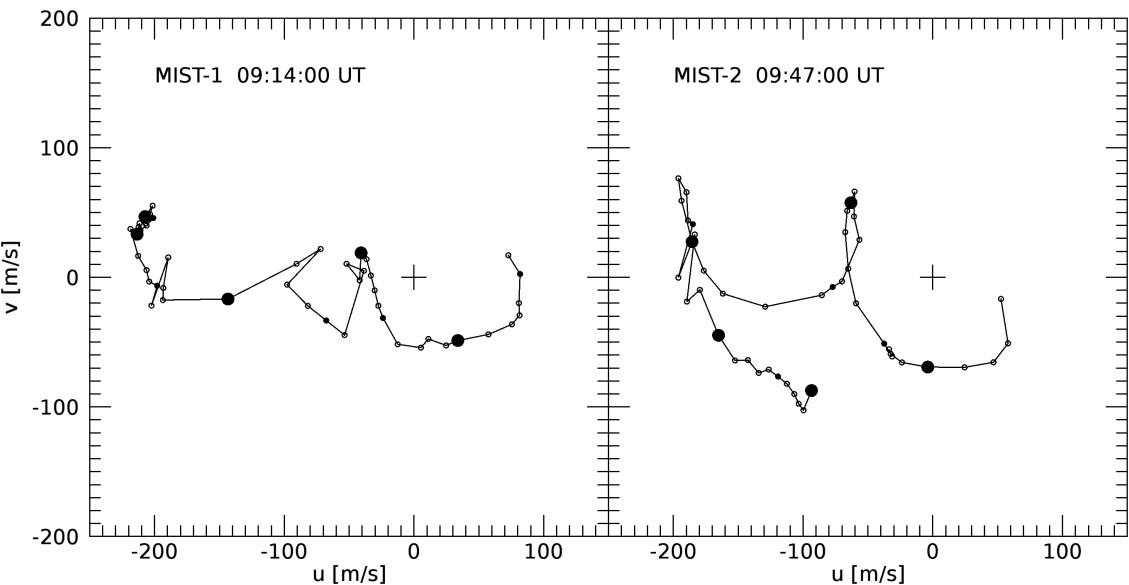

**Figure 14.** Hodograph projections of the horizontal wind. The lowest altitudes start on the right at positive zonal winds. Small open circles are drawn every 1 km and large filled circles every 10 km. The first large circle is at 90 km. The cross marks the origin (zero wind).





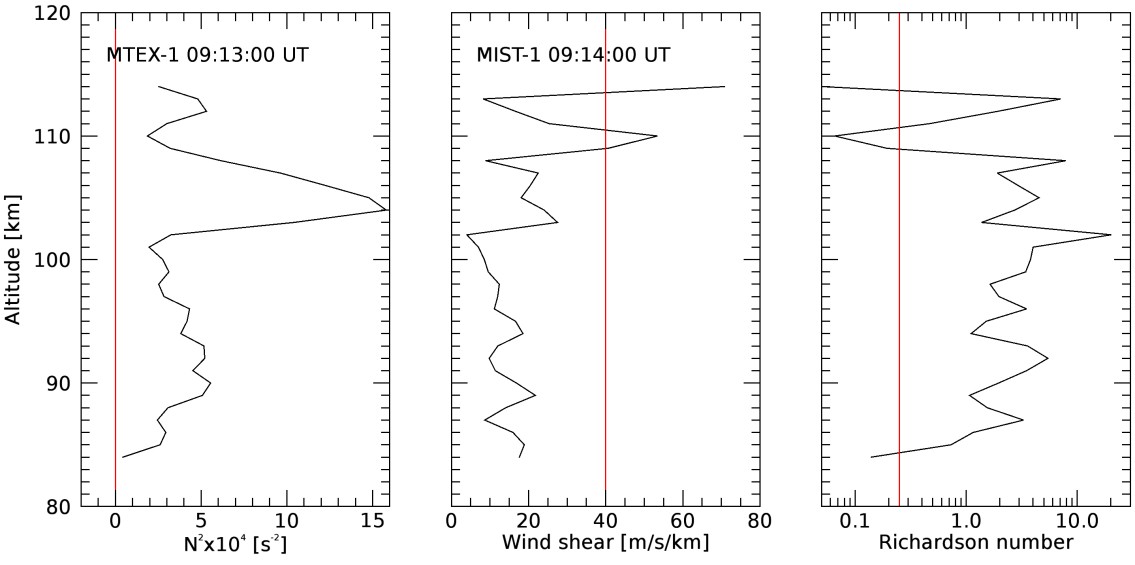

**Figure 15.** Buoyancy frequency from CONE upleg temperatures, horizontal wind shears from TMA winds, and Richardson numbers for the first salvo. The red line in the right panel indicates $Ri = 0.25$.



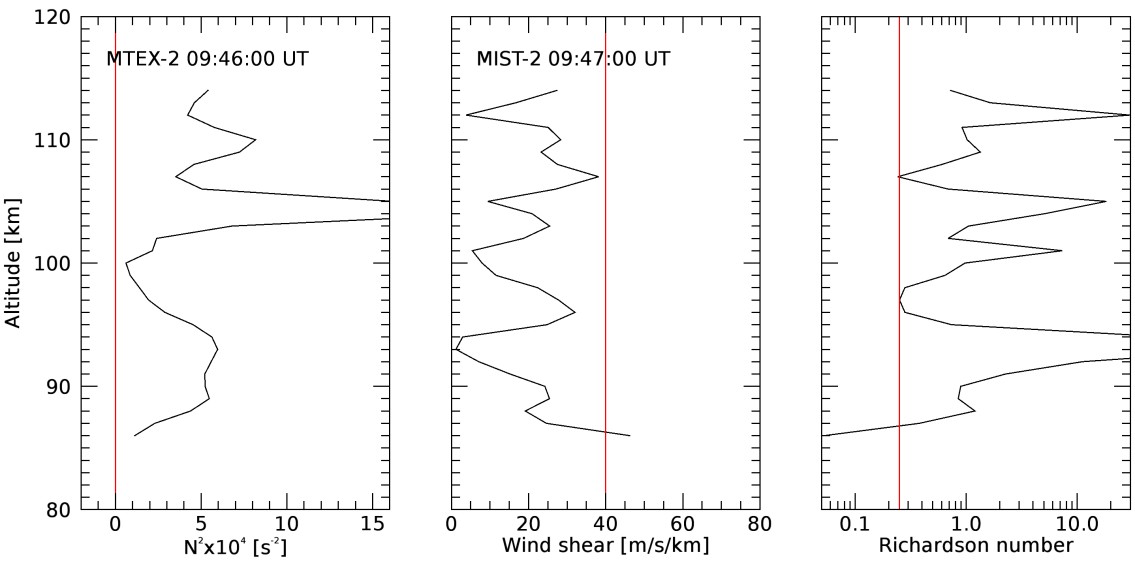

**Figure 16.** Same as previous figure, but for second salvo.



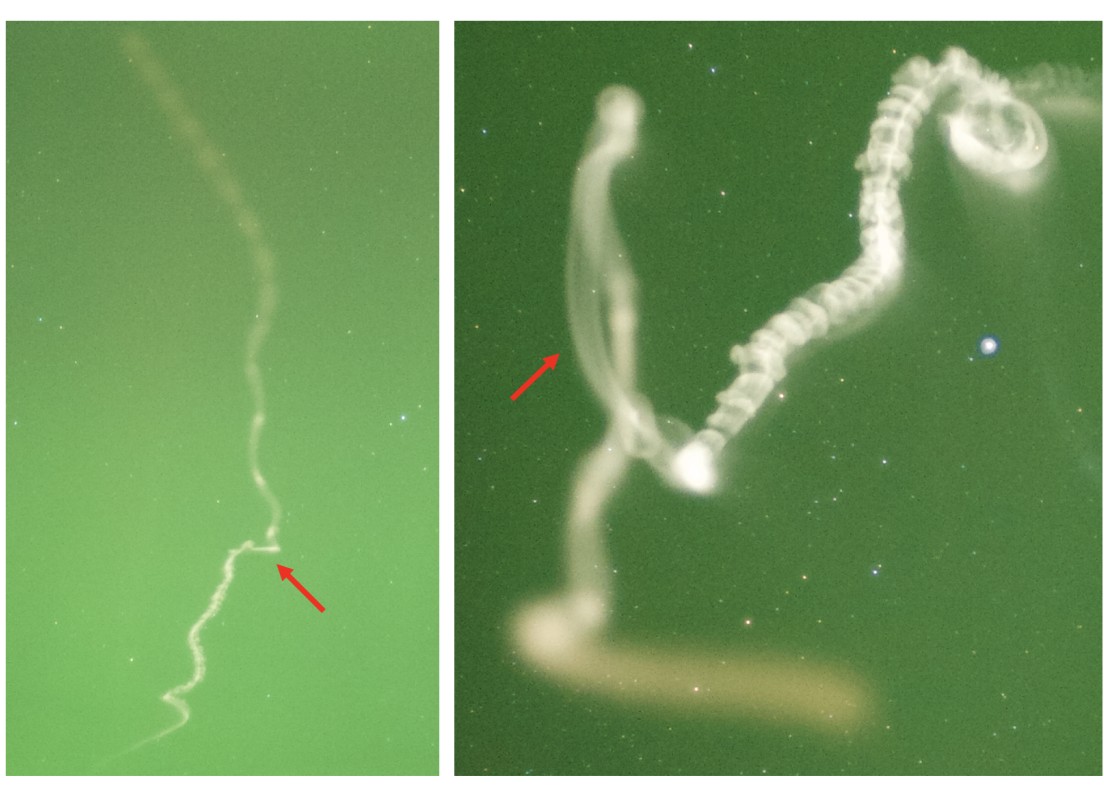

**Figure 17.** Photographs of the MIST-1 upleg trail from Toolik Lake (left) and Poker Flat (right) at 09:16:59 UT. The 110 km region is located where the trail is marked by the arrows.



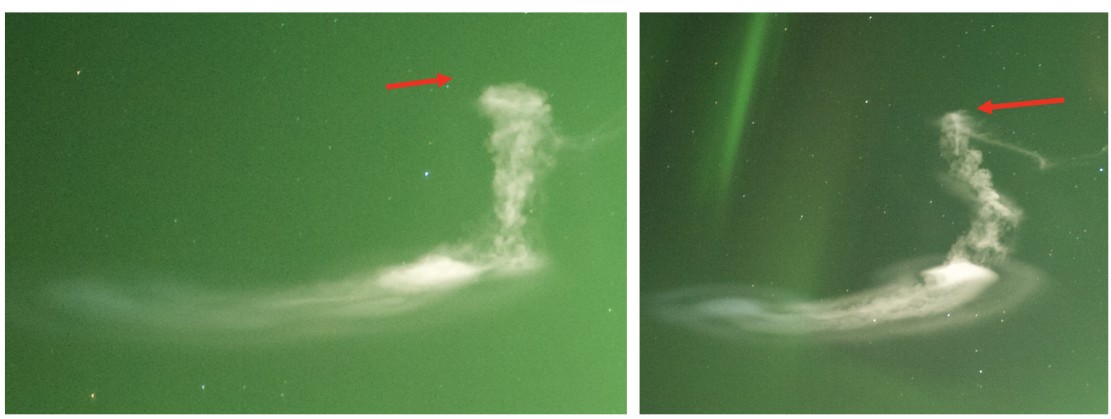

**Figure 18.** Photographs of the MIST-1 (left) and MIST-2 (right) downleg trail from Toolik Lake (left) taken at 09:23:24 and 09:56:24 UT, respectively. The arrows indicate the top of the quasi-adiabatic layer near 100 km.



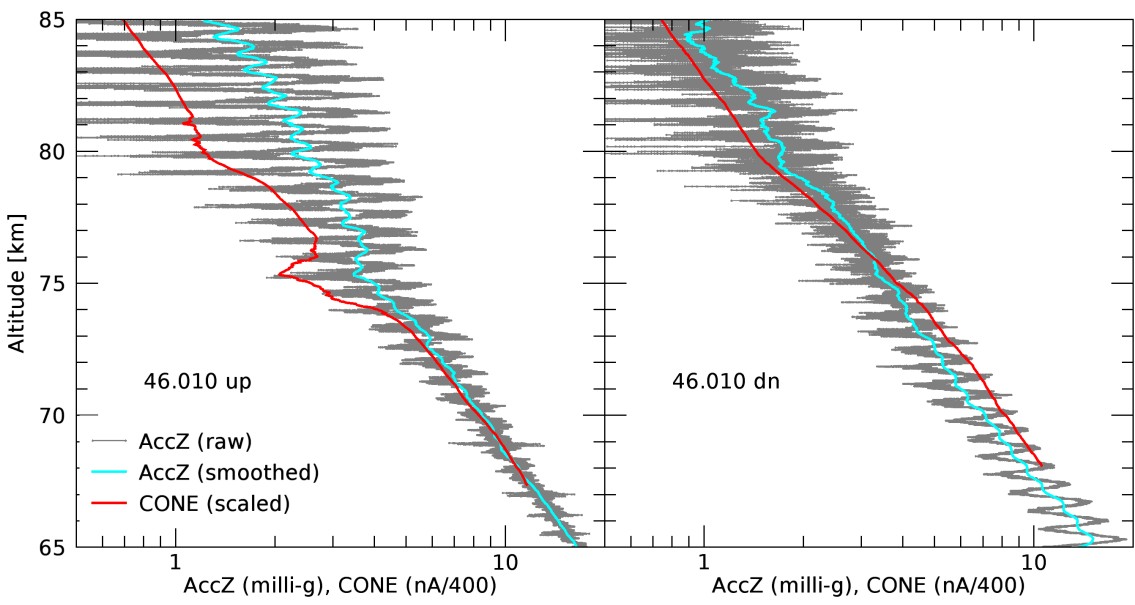

**Figure A1.** Accelerometer drag residual for flight 46.010 upleg (left) and downleg (right).