# Peer review of "On the short-term variability of turbulence and temperature in the winter mesosphere"

_Annales Geophysicae, 2018_

## Referee Comment (RC1) · Anonymous Referee #1 · 18 Jun 2018

On the short term variability of turbulence and temperature in the winter mesosphere

ByG.A.Lehmacher, M.F.Larsen, R.L.Collins, A.Barjatya, and B.Strelnikov

General comments

1The paper is one in a group of four concerning small scale dynamics in the mesosphere and lower thermosphere.

2The present paper discusses fluctuations and turbulence as measured during four rocket flights at high latitudes.

3The experiments described were carefully designed and successfully performed.

4The results show an intersting relationship of turbulence intensity and the gradient of

the vertical temperature profile.

5The turbulence data are highly variable, but are within the scope of earlier measurements.

6The paper is well written, and most ofthe pictures are carefully designed.

7The paper is recommended for publication after some improvements have been made.

Specific comments

1Page 11, lines 23, 26, and several other places:PhD theses of Szewczyk (2015) and Triplett (2016) are extensively used for comparison and discussion. This is not helpfulas that work is not normally available to the reader. Please give recent citations in the general literature.

2Please explain if the TMA experiment was anyhow affected by the bright auroral arc present simultaneously, or why not.

3Section 2.2:Please give a sketch of the CONE instrument (ion source).

4Figure 7:This picture is difficult to read!a) The profiles "46.009 upleg" and "SABER 64N230W..."

are almost not readable. b) The colour of the dotted line (SABER) is said to be magenta, but this is not

visible. c) Page 7, line 20:"The gaps are shown with dashed lines" :Very difficult to see..

5Figure 18:a) Please add an arrow at the 95 km altitude. b)Please explain from where these profiles

are seen. c) Please explain the stretched parts of the profiles in the lower parts of the pictures.

6.Page 12, Eq.(4):Please give a reference.

7.Page 13, l..30-32:Why is this seen only in the SABER data?

8.Page 14, l 11:Please explain"MEMS".

9Page 15, l 14: Pleaseadd the location of IAP.

10Page 15, l 15: Please add "Schwetzingen, Germany".

11Page 16, lines 30, 33;Please cite "Fritts,…2018a" and "Fritts….2018b".
* * *

---

## Referee Comment (RC2) · Anonymous Referee #2 · 29 Jun 2018

The authors present an in-depth analysis of a series of measurements taken with four sounding rockets launched in close temporal and spatial proximity. Two sounding rockets carried ionization gauges to measure neutral density profiles (from which temperatures can be inferred) and small scale neutral density fluctuations from which the turbulent energy dissipation rate was determined. The other two sounding rockets carried TMA payloads for measuring the wind field at high spatial resolution. The in-situ measurements were accompanied by ground based sodium lidar measurements from which temperature profiles were derived.

This is a unique set of measurement that allows a first experimental insight into the spatial and temporal variability of turbulence in the mesosphere. While numerical simulations have advanced to the point of making predictions of the likely morphology

and temporal variability of small scale flows in the mesosphere, the corresponding experimental data base is extremely rare - if not absent. As such, this is an important contribution to the literature that certainly warrants publication after some improvements have been made. As such my recommendation is to accept this manuscript for publication subject to minor revisions.

Comments:

- When formulating the scientific objective of their study, the authors state that they intend to "provide a detailed discussion on the static stability and turbulence structure for each profile following the methods developed for neutral density measurements in the mesosphere and lower thermosphere". With all due respect, this is not a scientific objective! Please formulate the scientific objective properly (spatial and temporal variability)and then also come back to this in the conclusions and abstract of this article.

- Page 2, line 4: I suggest to add a reference to the paper by Lübken and von Zahn, JGR 1991.

- Page 2, line 7/8: the general increase of energy dissipation and eddy diffusion with what? with altitude?

- Page 5, lines 31-33: I haven't understood how this works; the thesis of Triplett is not available to me. Please explain in more detail - or delete, since it doesn't matter.

- Page 6, line 6: Typical inner scales... Add: in the mesosphere and lower thermosphere

- Page 7, line 20: The first half sentence sounds rather poetic, maybe re-formulate in a more scientific style.

- Page 7, line 23: "The thermosphere is unusually structured" - compared to which other measurements/data base. Please provide reference.

- Page 7, line 27: Please indicate the distance between the SABER tangent points and

the in-situ measurements

- Page 7, line 29: I suggest to move the internet source of the SABER data to the "data availability" section in after the main manuscript text. See instructions of the publisher.

- Page 8, line 8: Please also show the lidar temperature profile.

- Page 8, line 13/14: I do not think that this is an appropriate way to estimate the absolute temperature error. It does describe the difference between two measurements - OK. But one is a nightly mean and the other a snapshot. I recommend to remove this from the figures and just add a general sentence about the difference between the different measurement (lidar, in situ).

- Page 8, line 17/18: The wording is sloppy here: $N^2=0$ corresponds to an adiabatic lapse rate but is not the same. Also, the value for stable conditions is completely arbitrary. Please reformulate.

- Page 9, lines 30-31: Is it possible to summarize the observed morphology in a schematic drawing? This will maybe also make it easier to compare to simulations by Fritts et al. and extract the scientific content of this study.

- Page 10, line 9: Wouldn't it be more instructive to first remove the shear or large scale background and then show the hodographs? This would make it easier for the reader to recognize wave features.

- Page 10, line 13: Here and in a few other places the authors compare their findings to results from an earlier rocket flight. However, they leave the reader with the question what they should learn from the comparison. I suggest to either discuss this comparison in more depth and draw a conclusion or to delete it.

- Page 10, line 15: The Richardson number has been used as an index for instability already since the work by Miles and Howard, both in 1961:

Miles, J.W. On the stability of heterogeneous shear flows. J. Fluid Mech. 10, 496–508,

1961.

Howard, L.N. Note on a paper of John W. Miles. J. Fluid Mech. 10, 509– 512, 1961.

- Page 10, line 15: Well, but the epsilon-measurements only give a "zero measurement" in the altitude range of overlap. This should be acknowledged.

- Page 10, line 19-23: At some point the authors should clearly say that they have no coincident measurements of epsilon and the other parameters discussed here (at least no values different from their detection threshold).

- Page 10, line 34: Please explain why. What should be driving the convection here?

- Page 11, line 12: Can this be formulated in a more quantitative manner? What kind of impact? How large?

- Page 12, line 26: "not or not breaking" - something is missing here.

- Page 12, lines 27 - 30: When mentioning the results of Achatz (2007): what is the conclusion for the current work beyond mentioning that these theoretical results exist?

- Page 13, line 1: Do you really mean deep convection? Wouldn't you then need to present vertical velocities? I suggest to reformulate this.

- Page 13, last sentence: either delete or be more specific. Otherwise the statement is trivial.

---

## Author Comment (AC1) · 6 Jul 2018

***Manuscript*** **"On the short-term variability of turbulence and temperature in the winter mesosphere"** ***by*** **Gerald A. Lehmacher et al.**

**Author's Response to**

**Anonymous Referee #1**

*General comments*

*1 The paper is one in a group of four concerning small scale dynamics in the mesosphere and lower thermosphere.*

*2 The present paper discusses fluctuations and turbulence as measured during four rocket flights at high latitudes.*

*3 The experiments described were carefully designed and successfully performed.*

*4 The results show an interesting relationship of turbulence intensity and the gradient of the vertical temperature profile.*

*5 The turbulence data are highly variable, but are within the scope of earlier measurements.*

*6 The paper is well written, and most of the pictures are carefully designed.*

*7 The paper is recommended for publication after some improvements have been made.*

Author's Response: We appreciate the general comments. Our responses are below.

*Specific comments*

*1. Page 11, lines 23, 26, and several other places: PhD theses of Szewczyk (2015) and Triplett (2016) are extensively used for comparison and discussion. This is not helpful as that work is not normally available to the reader. Please give recent citations in the general literature.*

Author's Response: The work in Triplett (2016) will now be referenced as Triplett et al. (2018), "Observations of reduced turbulence and wave activity in the Arctic middle atmosphere following the January 2015 stratospheric sudden warming", J. Geophys. Res. Atmospheres. This publication was still under review a few weeks ago, but publication is expected in the next week.

The PhD thesis by Szewczyk (2015) is public and easily accessible at http://rosdok.uni-rostock.de/site/epub and we will include this link. A manuscript covering similar results is in preparation (B. Strelnikov, private communication), however, at this point cannot be cited. We prefer to keep the Szewczyk (2015) reference, since it contains updated turbulence statistics.

Changes in Manuscript: Add additional reference Triplett et al. (2018) and web link http://rosdok.uni-rostock.de/site/epub

*2. Please explain if the TMA experiment was anyhow affected by the bright auroral arc present simultaneously, or why not.*

Author's Response: The photography of the TMA trails may be affected by a larger sky background brightness under aurora. This may be important when analyzing the structure function as done in Roberts and Larsen (2014). The wind measurements are less affected since the large-scale features of the trails are usually well visible over the background. The auroral brightness changes between images and some trail may have a larger position uncertainty than others. Given the large number of images and positions for a wind determination, the typical wind error (5-10 m/s) remains the same.

Changes in Manuscript: None

*3. Section 2.2: Please give a sketch of the CONE instrument (ion source).*

Author's Response: A sketch of the CONE instrument can be found in Rapp et al. (2001).

Changes in Manuscript: We will add: A sketch of the CONE instrument can be found in Rapp et al. (2001).

*4. Figure 7: This picture is difficult to read!*

*a) The profiles "46.009 upleg" and "SABER 64N230W. . ." are almost not readable.*

*b) The colour of the dotted line (SABER) is said to be magenta, but this is not visible.*

*c) Page 7, line 20:"The gaps are shown with dashed lines" :Very difficult to see…*

Author's Response: (a,b) We will improve the readability of Fig. 7. Add lines to the legend, pick different color for SABER 64N 230W. (c) We will add short vertical lines indicating the dashed regions. Note that all 4 profiles are shown individually in Figures 9-12.

Changes in Manuscript: We will modify Fig.7 for better readability, mainly the pale-pink SABER profile.

*5. Figure 18:*

*a) Please add an arrow at the 95 km altitude.*

*b) Please explain from where these profiles are seen.*

*c) Please explain the stretched parts of the profiles in the lower parts of the pictures.*

Author's response: (a) Yes.

(b) Will explain that profiles are seen from North and give coordinate Toolik Lake (68.63 N, 149.60 W).

(c) Will explain that lower part of trail is stretched to the left due to predominantly eastward winds.

Changes in Manuscript: Will add changes to Figure 18 and explanations.

*6. Page 12, Eq.(4): Please give a reference.*

Author's Response: Will include Lübken (1992), On the extraction of turbulent parameters from atmospheric density fluctuations, J. Geophys. Res., 97, 20,385-20,395.

Changes in Manuscript: Include above reference.

*7. Page 13, l..30-32: Why is this seen only in the SABER data?*

Author's Response: There is a parenthesis missing after "MIL type".

Meriwether and Gerrard (2003) write about MILs: "This phenomenon occurs quite often, especially in the midlatitude winter hemisphere, may last for many days, and is observed to have a broad horizontal distribution thousands of kilometers in scale."

It is my view that a MIL should be spatially extended and also last for a couple of hours. This is difficult to observe. The statement about the possibility of a MIL near 95 km is only based on the two SABER profiles that were observed hours and 100s of km apart from Poker Flat. The Rayleigh lidar could not provide data at these altitudes, as lidar data are often used to diagnose MILs.

Overall, the parenthetical comment on a MIL near 95 km is speculative and will be removed.

Changes in Manuscript: Will remove comment.

*8. Page 14, l 11: Please explain "MEMS".*

Author's Response: Microelectromechanical System

Changes in Manuscript: Add explanation.

*9. Page 15, l 14: Please add the location of IAP.*

Changes in Manuscript: Will add Kühlungsborn, Germany.

*10. Page 15, l 15: Please add "Schwetzingen, Germany".*

Changes in Manuscript: Add Schwetzingen, Germany.

*11. Page 16, lines 30, 33: Please cite "Fritts,. . .2018a" and "Fritts. . ..2018b".*

Author's Response:  a and b is missing in References

Changes in Manuscript: Will correct references.

---

## Author Comment (AC2) · 6 Jul 2018

*Manuscript* **"On the short-term variability of turbulence and temperature in the winter mesosphere"** *by* **Gerald A. Lehmacher et al.**

**Author's Response to**

**Anonymous Referee #2**

*The authors present an in-depth analysis of a series of measurements taken with four sounding rockets launched in close temporal and spatial proximity. Two sounding rockets carried ionization gauges to measure neutral density profiles (from which temperatures can be inferred) and small scale neutral density fluctuations from which the turbulent energy dissipation rate was determined. The other two sounding rockets carried TMA payloads for measuring the wind field at high spatial resolution. The in-situ measurements were accompanied by ground based sodium lidar measurements from which temperature profiles were derived.*

*This is a unique set of measurement that allows a first experimental insight into the spatial and temporal variability of turbulence in the mesosphere. While numerical simulations have advanced to the point of making predictions of the likely morphology and temporal variability of small scale flows in the mesosphere, the corresponding experimental data base is extremely rare - if not absent. As such, this is an important contribution to the literature that certainly warrants publication after some improvements have been made. As such my recommendation is to accept this manuscript for publication subject to minor revisions.*

Author's Response: We appreciate the Referee's comments. Our responses are listed below.

*Comments:*

*- When formulating the scientific objective of their study, the authors state that they intend to "provide a detailed discussion on the static stability and turbulence structure for each profile following the methods developed for neutral density measurements in the mesosphere and lower thermosphere". With all due respect, this is not a scientific objective! Please formulate the scientific objective properly (spatial and temporal variability) and then also come back to this in the conclusions and abstract of this article.*

Author's Response: Thank you for the comment. The purpose of this paper is to examine the spatial and temporal variability of mesospheric turbulence in relationship to the static stability of the background atmosphere.

Changes in Manuscript: Will reformulate purpose of paper and add objective to abstract and conclusions at appropriate places.

*- Page 2, line 4: I suggest to add a reference to the paper by Lübken and von Zahn, JGR 1991.*

Author's Response: Yes, this makes sense to add this reference on line 4.

Changes in Manuscript: Add reference.

*- Page 2, line 7/8: the general increase of energy dissipation and eddy diffusion with what? with altitude?*

Author's Response: "with altitude".

Changes in Manuscript: Add "with altitude"

*- Page 5, lines 31-33: I haven't understood how this works; the thesis of Triplett is not available to me. Please explain in more detail - or delete, since it doesn't matter.*

Author's Response: Some of the references to Triplett (2016) will be replaced by Triplett et al. (JGR, 2018), which was under review, but will be published in this month. Indeed, it does not matter for the results of this paper, but for a reader who will closely compare both papers, this explanation will make sense. PS. The thesis is available at Scholarworks.alaska.edu

Changes in Manuscript: Will add new reference and revise explanation as needed.

*- Page 6, line 6: Typical inner scales... Add: in the mesosphere and lower thermosphere*

Author's Response: OK.

Changes in Manuscript: Will add.

*- Page 7, line 20: The first half sentence sounds rather poetic, maybe reformulate in a more scientific style.*

Author's Response:

*- Page 7, line 23: "The thermosphere is unusually structured" - compared to which other measurements/data base. Please provide reference.*

Author's Response: This is a good catch. Compared with many in situ observations of temperature between 100 and 110 km by mass spectrometers and ionization gauges (e.g. individual profiles for Lübken and von Zahn, 1991), the structuring is perhaps not unusual. Will replace with "highly structured"

Changes in Manuscript: Change to "highly structured"

*- Page 7, line 27: Please indicate the distance between the SABER tangent points and the in-situ measurements*

Author's Response: distances are approximately 840 and 310 km, which is large compared to rocket profile separation

Changes in Manuscript: Add distances.

*- Page 7, line 29: I suggest to move the internet source of the SABER data to the "data availability" section in after the main manuscript text. See instructions of the publisher.*

Author's Response: OK.

Changes in Manuscript: Move.

*- Page 8, line 8: Please also show the lidar temperature profile.*

Author's Response: Will include 120-min averaged lidar profile for time around launch (up to about 85 km). This will show the described features. More lidar data and wave activity can be found in paper by Triplett et al (2018), which will be referenced.

Changes in Manuscript: Include lidar profile.

*- Page 8, line 13/14: I do not think that this is an appropriate way to estimate the absolute temperature error. It does describe the difference between two measurements - OK. But one is a nightly mean and the other a snapshot. I recommend to remove this from the figures and just add a general sentence about the difference between the different measurement (lidar, in situ).*

Author's Response: Both temperature profiles are based on the CONE in situ data. However, it is true that the calibrated densities and ram correction is the standard and appropriate way to calculate temperatures. Therefore, I will omit the other profiles. It will not change the discussion of the profiles or conclusions.

Changes in Manuscript: Change plots of individual temperature and BV frequency profiles.

*- Page 8, line 17/18: The wording is sloppy here: $N^2=0$ corresponds to an adiabatic lapse rate but is not the same. Also, the value for stable conditions is completely arbitrary. Please reformulate.*

Author's Response: Will reformulate. Green lines are included for comparisons with Fritts et al. (2018b)

Changes in Manuscript: Will reformulate.

*- Page 9, lines 30-31: Is it possible to summarize the observed morphology in a schematic drawing? This will maybe also make it easier to compare to simulations by Fritts et al. and extract the scientific content of this study.*

Author's Response: This is a fine idea to add a schematic of temperature profile and turbulence. Most turbulence layers are found in the stable portion of the middle mesosphere and very few in the near adiabatic region above. Since the paper contains already very many figures, I rather emphasize this result in the discussion and make comparison with Fritts et al. clearer.

Changes in Manuscript: Will make results clearer on page 9 and in 3rd paragraph of summary.

*- Page 10, line 9: Wouldn't it be more instructive to first remove the shear or large scale background and then show the hodographs? This would make it easier for the reader to recognize wave features.*

Author's Response: This is a good idea for extracting wave parameters. However, the hodographs including the full wind vector help in understanding the trail images at the end of the paper.

Changes in Manuscript:

*- Page 10, line 13: Here and in a few other places the authors compare their findings to results from an earlier rocket flight. However, they leave the reader with the question what they should learn from the comparison. I suggest to either discuss this comparison in more depth and draw a conclusion or to delete it.*

Author's Response: We want to point out the existence of large-scale wave activity both under quiet (2009) and active (2015) conditions. This can be deduced from the wind measurements, but not from the temperature profiles alone. In addition, previous measurements (cited) show the connection between strong wind shear and auroral forcing. Our results also indicate perturbed temperatures under such conditions. This is the conclusion of this comparison. There are various mechanisms possible for such heating, but without additional measurements and modeling, they remain speculation and should not be added.

Changes in Manuscript: Will add depth to the discussion.

*- Page 10, line 15: The Richardson number has been used as an index for instability already since the work by Miles and Howard, both in 1961:*

Miles, J.W. On the stability of heterogeneous shear flows. J. Fluid Mech. 10, 496–508, 1961.

Howard, L.N. Note on a paper of John W. Miles. J. Fluid Mech. 10, 509– 512, 1961.

Author's Response: Thank you. We will include the references.

Changes in Manuscript: Include references.

*- Page 10, line 15: Well, but the epsilon-measurements only give a "zero measurement" in the altitude range of overlap. This should be acknowledged.*

Author's Response: That is true in this case. Will be acknowledged.

Changes in Manuscript: Mention non-existence of small-scale turbulence in overlap region.

*- Page 10, line 19-23: At some point the authors should clearly say that they have no coincident measurements of epsilon and the other parameters discussed here (at least no values different from their detection threshold).*

Author's Response: We will add such a statement.

Changes in Manuscript: We do not have other measurements of epsilon in region of lower thermosphere.

*- Page 10, line 34: Please explain why. What should be driving the convection here?*

Author's Response: The region of adiabatic gradients may include super-adiabatic conditions and cause the acceleration and large vertical displacements of air parcels. According to modeling, such regions are not subject of strong wave breaking and turbulence generation, since $N^2 \sim 0$ and medium scale waves, which are important for turbulence, are evanescent in these regions (J. Snively, pers. communication, 2018)

Changes in Manuscript: We will add more explanation to our observation.

*- Page 11, line 12: Can this be formulated in a more quantitative manner? What kind of impact? How large?*

Author's Response: I carefully re-read the conclusions formulated in Fritts et al. (2018b, Section 6) and which are summarized on Page 11, line 12. A quantitative formulation of these specific simulation results is not easy to extract, since the model results also represent "snapshots" of a complex environment (Fritts et al., 2018b). A good summary would be that a "plausible range" of the Prandtl number under these conditions is 2-4. For the scope of this paper, the sentence starting with "However," can be deleted, since we do not measure heat flux or observe detailed gravity wave breaking or evidence for KHI.

Changes in Manuscript: delete sentence

*- Page 12, line 26: "not or not breaking" - something is missing here.*

Author's Response: It should read: "that are overturning, either partially or fully, but not breaking".

Changes in Manuscript: change sentence

*- Page 12, lines 27 - 30: When mentioning the results of Achatz (2007): what is the conclusion for the current work beyond mentioning that these theoretical results exist?*

Author's Response: The theoretical work is mentioned here, because it may be applicable to the relatively quiet layer between 82 and 87 km subject to gravity wave and sodium layer overturning. However, it may not be well connected to the available data and can therefore be removed.

Changes in Manuscript: Remove results from Achatz.

*- Page 13, line 1: Do you really mean deep convection? Wouldn't you then need to present vertical velocities? I suggest to reformulate this.*

Author's Response: Strike "deep". See also Comment for Page 10, line 34. This very interesting sequence of trail images (which may reveal also vertical velocities) will be analyzed in more detail for a future publication.

Changes in Manuscript: Strike "deep".

*- Page 13, last sentence: either delete or be more specific. Otherwise the statement is trivial.*

Changes in Manuscript: Delete.